# Identification of AURKA as a Biomarker Associated with Cuproptosis and Ferroptosis in HNSCC

**DOI:** 10.3390/ijms25084372

**Published:** 2024-04-16

**Authors:** Xiao Jia, Jiao Tian, Yueyue Fu, Yiqi Wang, Yang Yang, Mengzhou Zhang, Cheng Yang, Yijin Liu

**Affiliations:** 1State Key Laboratory of Medicinal Chemical Biology, College of Pharmacy and Tianjin Key Laboratory of Molecular Drug Research, Nankai University, Tianjin 300000, China; 1120210643@mail.nankai.edu.cn (X.J.); 1120210647@mail.nankai.edu.cn (J.T.); 2120211398@mail.nankai.edu.cn (Y.F.); 2120221665@mail.nankai.edu.cn (Y.W.); 2120221670@mail.nankai.edu.cn (Y.Y.); 2Key Laboratory of Evidence Science, China University of Political Science and Law University, Beijing 100088, China; 3Collaborative Innovation Center of Judicial Civilization, China University of Political Science and Law, Beijing 100088, China

**Keywords:** head and neck squamous cell carcinoma, prognostic model, cuproptosis, ferroptosis, AURKA

## Abstract

Cuproptosis and ferroptosis represent copper- and iron-dependent forms of cell death, respectively, and both are known to play pivotal roles in head and neck squamous cell carcinoma (HNSCC). However, few studies have explored the prognostic signatures related to cuproptosis and ferroptosis in HNSCC. Our objective was to construct a prognostic model based on genes associated with cuproptosis and ferroptosis. We randomly assigned 502 HSNCC samples from The Cancer Genome Atlas (TCGA) into training and testing sets. Pearson correlation analysis was utilized to identify cuproptosis-associated ferroptosis genes in the training set. Cox proportional hazards (COX) regression and least absolute shrinkage operator (LASSO) were employed to construct the prognostic model. The performance of the prognostic model was internally validated using single-factor COX regression, multifactor COX regression, Kaplan–Meier analysis, principal component analysis (PCA), and receiver operating curve (ROC) analysis. Additionally, we obtained 97 samples from the Gene Expression Omnibus (GEO) database for external validation. The constructed model, based on 12 cuproptosis-associated ferroptosis genes, proved to be an independent predictor of HNSCC prognosis. Among these genes, the increased expression of aurora kinase A (AURKA) has been implicated in various cancers. To further investigate, we employed small interfering RNAs (siRNAs) to knock down AURKA expression and conducted functional experiments. The results demonstrated that AURKA knockdown significantly inhibited the proliferation and migration of HNSCC cells (Cal27 and CNE2). Therefore, AURKA may serve as a potential biomarker in HNSCC.

## 1. Introduction

Head and neck squamous cell carcinoma (HNSCC) is a prevalent form of cancer globally, found to develop from the mucosal epithelium of the mouth, pharynx, and larynx. It is the most prevalent malignancy of the head and neck region [1,2]. Currently, the primary treatment options for HNSCC include surgery, chemotherapy, radiotherapy, and primary chimeric antigen receptor T cell immunotherapy (CRAT). However, 50% of patients have a poor prognosis at advanced stages [3]. It has been suggested that incorporating prognostic and predictive signatures into clinical management may be a solution to overcoming barriers to targeted cancer therapy and improving patient survival. Therefore, there is an urgent need to identify signatures to provide new interventions for treating HNSCC.

Metal ions are crucial micronutrients in maintaining cellular homeostasis, regulating metabolic pathways, facilitating signaling and energy conversion, and other activities. However, these ions’ inadequate or excessive distribution can significantly affect various physiological functions, leading to apoptosis [4]. For instance, ferroptosis is an iron-dependent form of apoptosis. An excess accumulation of intracellular iron results in lipid peroxide accumulation, leading to programmed cell death in tumor cell membranes [5,6]. HNSCC cells usually exhibit increased intracellular iron concentrations, and ferroptosis-inducing therapy effectively induces apoptosis in HNSCC [7]. Similarly, cuproptosis is a copper-dependent mode of cell death. It has been reported that cuproptosis is predominantly caused by the irregular accumulation of lipid-acylated proteins and the depletion of iron–sulfur cluster proteins. Combining copper ions with the lipid-acylated components of the tricarboxylic acid cycle triggers a toxic stress response to the proteins, leading to cell death [8,9]. It was demonstrated that inhibition of cuproptosis could increase HNSCC cell viability in patients with oral squamous cell carcinoma [8]. Recent research has suggested an underlying connection between ferroptosis and cuproptosis. For example, in hepatocellular carcinoma, the ferroptosis inducers sorafenib and erastin augmented copper death induced by copper ion carriers in cancer by upregulating protein lipid acylation and inhibiting intracellular glutathione (GSH) synthesis [10]. In lung cancer, Bennett, W.D. et al. constructed a diagnostic model based on copper apoptosis/ferroptosis genes that can be used to identify lung adenocarcinoma (LUAD) patients suitable for immunotherapy and predict sensitivity to chemotherapeutic agents. Similarly [11], Garcia Morillo et al. screened for copper and ferroptosis genes that can predict the survival of breast cancer (BRCA) patients by using bioinformatics and constructed a risk regression model, which can effectively predict the prognosis of breast cancer patients and enable the personalized treatment of patients [12]. Yet, the relationship between cuproptosis combined with ferroptosis-related genes and the prognosis of HNSCC is not fully understood.

Recent evidence suggests that combining the two predictive signatures improves the accuracy of the predictive model and provides new ideas for constructing clinical prognostic tools for various cancers [13]. For instance, such a novel model based on genes related to cuproptosis and ferroptosis possessed excellent potential for predicting prognosis in lung adenocarcinoma [14], ovarian cancer [15], and hepatocellular carcinoma [16]. Cuproptosis and ferroptosis have a notable impact on the prognosis of HNSCC. However, few studies have focused on the prognostic signatures associated with cuproptosis and ferroptosis in HNSCC. Therefore, this study aims to construct a prognostic model for HNSCC based on cuproptosis- and ferroptosis-related genes, predicting the prognosis, immune characteristics, and potential drugs for HNSCC.

In this study, we aimed to construct a predictive prognostic model based on cuproptosis-associated ferroptosis genes in HNSCC. We found that, of the cuproptosis-associated ferroptosis genes modeled, aurora kinase A (AURKA) was upregulated in HNSCC patients with tumor suppressor p53 (TP53) mutations and human papillomavirus uninfected (HPV), and could be associated with poor prognosis [17]. Therefore, we investigated the potential prognostic value of AURKA by using functional assays. Our study’s results may help improve individualized treatment and prognostic assessment of HNSCC.

## 2. Results

### 2.1. Construction of the Prognostic Model in HNSCC

To investigate cuproptosis-associated ferroptosis genes in HNSCC, we first explored the correlation network of cuproptosis genes themselves (Figure 1A). The results revealed significant correlations between dihydrolipoamide s-acetyltransferase (DLAT), dialectical behavior therapy (DBT), metal regulatory transcription factor 1 (MTF1), nonalcoholic fatty liver disease (NFE2L2), and developmental language disorder (DLD). This suggests that cuproptosis plays an essential role in HNSCC. Next, through the Pearson correlation analysis, cuproptosis-associated ferroptosis genes in HNSCC were obtained (Appendix A). We then performed cluster analysis on the expression of these cuproptosis-associated ferroptosis genes in HNSCC patients (Figure 1B). Through univariate Cox proportional hazards (COX) regression analysis, we screened 30 cuproptosis-associated ferroptosis genes with prognostic significance (Figure 1C). Predictive features based on prognostic-related genes were further analyzed using LASSO, Cox regression to minimize possible overfitting issues and identify optimal criteria. Twelve cuproptosis-associated ferroptosis genes were screened out on the basis of the optimal penalty parameter λ determined by tenfold cross-validation following the minimum criteria. They were used to construct the prognostic model, which can be used to identify the most effective predictive signature and generate prognostic indicators to predict clinical outcomes (Figure 1D,E).

We constructed the risk prognosis model for HNSCC to assess the predictive risk of patients based on the 12 cuproptosis-associated ferroptosis genes. HNSCC samples were classified into high- and low-risk groups, using the median risk values as a predictive signature. The risk class distribution, expression levels of cuproptosis-associated ferroptosis genes, patient survival status, and time pattern results of The Cancer Genome Atlas (TCGA) training set are shown in Figure 2A,B. In addition, we assessed the overall survival of patients in the HNSCC training set by Kaplan–Meier (KM) analysis and found that individuals in the high-risk category had a considerably worse prognosis compared to those in the low-risk group (Figure 2C). Figure 2D demonstrates the receiver operating characteristic (ROC) prediction performance of our constructed risk prognostic model, and it can be found that our model can successfully predict the 1-, 3-, and 5-year survival of patients with statistical significance. Figure 2E demonstrates the heatmap of the expression of the 12 cuproptosis-associated ferroptosis genes in the training set. Furthermore, we explored the ability of the prognostic model to differentiate patients using a principal component analysis (PCA) approach with the 12 cuproptosis-associated ferroptosis genes as indicators. The results showed that the risk prognostic model we constructed was able to accurately differentiate patients into two groups, high-risk and low-risk (Figure 2F).

### 2.2. Evaluation and Validation of the Prognostic Model in HNSCC

To evaluate the risk prediction model constructed based on the cuproptosis-associated ferroptosis genes, we plotted the distribution of risk scores, overall patient survival, and ROC results for patients in the TCGA training set. Figure 3A,B show the risk class distribution of patients in the TCGA validation set and the KM analysis, and the results suggest that the higher the risk score of the patient, the higher the mortality rate. The ROC curve results suggest that the 1-year, 3-year, and 5-year predictions of this prognostic model are accurate and stable in the training set (Figure 3C). To validate the robustness and reliability of our constructed model in an independent cohort, we downloaded data for external validation from the GSE41613 dataset (platform: GPL570) uploaded previously by Lohavanichbutr, P. et al., which contains expression profiles and clinical information of 97 HPV-negative OSCCs [18]. We categorized the patients in the validation cohort into low-risk and high-risk groups based on the median risk score (Figure 3D). The Kaplan–Meier analysis results shown in Figure 3E affirm the continued statistical significance of the risk prognostic model within the GEO dataset. The ROC curves in Figure 3F confirm that the prognostic model performs well in assessing this prognosis of patients at 1, 2 and 3 years (all greater than 0.7). In conclusion, the above findings military confirm that the risk prognostic model we constructed is a good independent prognostic factor in HNSCC.

To further assess and validate whether the risk prognostic model we constructed is likely to be an independent prognostic factor, we further investigated the predictive value of this prognostic model using univariate and multivariate Cox regression analyses. The univariate COX analysis showed a hazard ratio (HR) of 2.3283 (95% confidence interval [CI]: 1.6957–3.1969, *p* < 0.001), and the multivariate analysis showed an HR of 2.1766 (95% CI: 1.5836–2.9917, *p* < 0.001, Figure 4A,B). These results suggested that the prognostic model may be an excellent independent prognostic factor for HNSCC patients. Additionally, we used the prognostic model to construct nomogram calibration plots to assess the 1-, 3-, and 5-year survival rates of HNSCC patients. The results suggest that the model is a reliable predictor of survival at one, three, and five years (Figure 4C,D).

### 2.3. The Relationship between Clinical Characteristics and the Prognostic Model

To investigate the prognostic significance of our constructed risk model under different clinicopathological characteristics, we randomly divided the patients into two subgroups according to clinical factors such as TNM staging, grading, age and gender. The findings in Figure 5A reveal that age, gender, and tumor stage were not independent prognostic factors. Still, the pathological stage was a reliable, independent prognostic factor in univariate and multifactorial COX regression analyses. The risk score did not statistically correlate with age, gender, and tumor stage, but it did exhibit some correlation with the TNM stage. Specifically, patients with T3/T4 tumors had higher risk scores than those with T1/T2 tumors, and patients with the N2/3 stage had higher risk scores than those with the N0/1 stage (Figure 5B). Using KM analysis, we evaluated the survival rates of high- and low-risk populations based on clinical characteristics and found that the survival rate of the high-risk group was significantly lower than that of the low-risk group. Female patients in the high-risk group had significantly lower survival rates than those in the low-risk group. Patients with stage IIIIV cancer had significantly lower survival rates than those in the low-risk group. Patients with N-stage 2/3 had significantly lower survival rates than those in the low-risk group. Patients with M0 stage had significantly lower survival rates than those in the low-risk group (Figure 5C).

### 2.4. Immunological Characteristics of the Prognostic Model

To further explore the immune characteristics of the population in our constructed risk prognostic model, we analyzed Tumor Immune Estimation Resource (TIMER) 2.0 and Translational Medicine Integrated Database (TISIDB) to determine whether tumor infiltrating immune cells (TIICs) and tumor-infiltrating lymphocytes (TILs) differed between the high-risk and low-risk groups to investigate the correlation between the prognostic model and immune infiltration [19,20]. As shown in Figure 6A, the infiltration of immune cells was different between the high-risk group and the low-risk group. Based on the genetic expression profile of tumor samples, along with the confirmed marker genes of several immune cells through studies, the single sample gene set enrichment analysis (ssGSEA) algorithm detected differences in some immune cells across different samples. Research has indicated that the immune cell infiltration of B cells and pulp cells plays a crucial synergistic role in inhibiting tumor development. Synergy promotes anti-tumor T cell immunity through its unique antigen, showing improved prognosis for patients [21]. Figure 6B shows that the activated CD8^+^ T cells and activated B cells of the lower risk group were higher than those of the higher risk group, which explains that the prognosis of the patients in the lower risk group was better than that of the patients in the higher risk group. Studies have shown that higher mesenchymal and immunological values in patients were associated with a good prognosis [21,22]. An immune checkpoint is a signal that regulates antigen recognition by T cell receptors during the immune response, and it may play an important role in carcinogenesis by promoting tumor immunosuppression. We explored the differential expression of immune checkpoints in the high- and low-risk groups, as shown in in Figure 6C–H. The results show that the expression of CD40 ligand (CD40LG) was significantly higher in the low-risk group than in the high-risk group (Figure 6C). The natural killer cell receptor 2B4 (CD244) was significantly higher in the low-risk group population than in the high-risk group population (Figure 6D). Benzodiazepine long-term administration (BLTA) was significantly higher in the low-risk group than in the high-risk group (Figure 6E). The cluster of differentiation 44 (CD44) was significantly lower in the low-risk group population than in the high-risk group population (Figure 6F). The expression of ADORO2A was significantly higher in the low-risk group than in the high-risk group (Figure 6G). The cluster of differentiation 27 (CD27) expression was significantly higher in the low-risk group than in the high-risk group (Figure 6H).

Furthermore, we used seven algorithms (TIMER, cell type identification by estimating relative subsets of RNA transcripts (CIBERSORT), CIBERSORTABS, QUANTISEQ, microenvironment cell populations-counter (MCPCOUNTER), XCELL, and the European prospective investigation into cancer and nutrition (EPIC)) to assess the immune function of the high- and low-risk groups. Our findings suggest that CD4^+^ T cells, B cells, and CD8^+^ T cells were functionally active in the low-risk group (Figure 7A). Our findings indicate that the high-risk cohort records lower scores compared to the low-risk cohort, suggesting that the high-risk group may receive an advantage from immunotherapy (Figure 7B). In addition, tumor mutational burden (TMB) is a commonly used biomarker to predict the immune checkpoint response in patients. Figure 7C,D shows the 15 genes commonly mutated in the high- and low-risk groups, with the 5 most frequently mutated genes in the high-risk populations being TP53, titin (TTN), FAT atypical cadherin 1 (FAT1), and cyclin-dependent kinase inhibitor 2A (CDKN2A). Additionally, we examined the immune infiltration function of the 12 cuproptosis-associated ferroptosis genes individually. We found that the expression of genes such as AURKA and glucose6phosphate dehydrogenase (G6PD) was negatively correlated with B cell and CD8^+^ T cell immune infiltration and positively correlated with tumor purity and CD4^+^ T cells (Appendix A). We further analyzed the TMB survival analyses of the high- and low-risk groups, as well as the survival differences between the high- and low-TMB and the high- and low-risk combination groups, and the results are shown in Figure 7E,F, where overall survival period (OS) was significantly better in the high-TMB group than in the low-TMB group (*p* < 0.05).

### 2.5. Functional Enrichment Analysis of the Prognostic Model

Enrichment analysis of genes from the high-risk and low-risk populations using the Gene ontology (GO)/Kyoto Encyclopedia of Genes and Genomes (KEGG) allowed for exploring risk-related biological functions and pathways. To further identify functional signaling pathways that might be enriched in the risk prediction model we constructed, we performed GO, KEGG, and Gene set enrichment analysis (GSEA) analyses. The results of the GO analysis indicated that biological processes (BPs) were mainly associated with the cellular response to chemical stress, response to oxidative stress, cellular response to oxidative stress, response to starvation, response to extracellular stimulus, cellular response to chemical stress, and response to starvation. The molecular functions (MFs) were mainly related to ubiquitin protein ligase binding, ubiquitin-like protein ligase binding, DNA-binding transcription factor binding, protein serine/threonine kinase activity, and oxidoreductase activity acting on nicotinamide adenine dinucleotide phosphate (NADPH). The cellular components (CCs) were mainly associated with the phagophore assembly site, autophagosome, phagophore assembly site membrane, protein kinase complex, and mitochondrial outer membrane (Figure 8A,B). The KEGG results suggest these genes are mainly enriched in autophagy, lipid and atherosclerosis, chemical carcinogenesis reactive oxygen species, FoxO signaling pathway, ferroptosis, central carbon metabolism in cancer, mitophagy, central carbon metabolism in cancer, Kaposi sarcoma-associated herpesvirus infection, and Kaposi sarcoma-associated pathways (Figure 8C,D, Appendix A).

The GSEA results suggested that ascorbate and alternate metabolism, pentose and glucuronate interconversions, drug metabolism, other enzymes, porphyrin metabolism, and soluble N-ethylmaleimide-sensitive factor attachment protein receptor (SNARE) interactions in vesicular transport pathways, were active in the high-risk population. In contrast, in the low-risk population, ATP-binding cassette (ABC) transporters, collecting duct acid secretion, fatty acid biosynthesis and degradation, thiamine metabolism pathways, and other pathways were active (Figure 8E,F).

### 2.6. Drug Sensitivity of the Prognostic Model

To investigate the potential dosing risks in the high-risk and low-risk populations, a selection of drugs was utilized to assess drug sensitivity. Appendix A shows that the Tivozanib, Vinorelbine, Tipifarnib, Thapsigargin, Talazoparib, Sorafenib, Salubrinal, Roscovitine, AKT inhibitor VIII, and other drugs had lower sensitivity in the high-risk population, whereas WZ184, THZ21021, and THZ249 had higher drug sensitivity in the high-risk population. These results confirm that cuproptosis-associated ferroptosis genes in HNSCC are correlated with drug sensitivity.

### 2.7. Identification of Potential Biomarkers in Constructed Model

To further validate the accuracy of the prognostic model and to explore potential biomarkers in the 12 genes in our constructed model, we further explored the expression of these genes in clinical samples. We know from the analysis of the results that among these 12 genes, 3 genes, AURKA, CAV1, and CDKN2A, have the most significant differential expression in HNSCC patients (*p* < 0.0001, Figure 9). Therefore, we selected these three genes to further explore their protein expression levels between HNSCC and normal subjects from the HPA database. Analysis of the results of the immunohistochemistry showed that the protein expression levels of AURKA, CAV1, and CDKN2A were all significantly different in HNSCC patients than in the normal group, which corroborated the results of the aforementioned RNA studies (Figure 9). All of the above findings further support our conclusion that the risk prognostic model we constructed may be an independent prognostic factor. Next, from other studies in the literature, we found that among these three genes, AURKA may be associated with poor prognosis in patients with a variety of cancers. The regulation of AUKRA may affect multiple signaling pathways in tumors, such as PI3K/Akt, mTOR, β-catenin/Wnt, and NF-κB pathways, and so on. However, studies on the prognostic mechanism of AURKA in HNSCC are unclear. Therefore, we selected the function of AUKRA in multiple HNSCC cells for further validation.

### 2.8. Inhibition of AURKA Suppresses the Proliferation and Migration of HNSCC Cells

To investigate the biological function of AURKA in HNSCC, we transfected two small interfering RNAs (siRNAs) to knock down their expression using Lipofectamine 2000 (Invitrogen, Waltham, MA, USA) and performed subsequent cell function experiments. The real-time reverse transcriptase polymerase chain reaction (RT-qPCR) results showed that siAURKA1 and siAURKA2 successfully reduced AURKA expression in CNE2 and Cal27 cells (Figure 10A,B). The Cell Counting Kit8 (CCK8) experiment (Yeasen Biotechnology, Shanghai, China) results showed that the inhibition of AURKA expression could inhibit the proliferative ability of Cal27 and CNE2 cells, with Cal27 cells exhibiting more significant inhibition (Figure 10C,D). Cell scratching experiments showed that the inhibition of AURKA significantly inhibited the migratory ability of HNSCC cells (Cal27 and CNE2 cell lines). The results demonstrate that the inhibition of AURKA expression can significantly inhibit the proliferation and migratory ability of HNSCC cells (Figure 10E,F).

## 3. Discussion

Currently, the primary treatment options for HNSCC include surgery, chemotherapy, radiotherapy, and CRAT. However, despite these treatments, 50% of patients experience a poor prognosis, particularly at advanced stages [23,24]. Emerging research suggests that cuproptosis and ferroptosis, two distinct forms of cell death, play crucial roles in the prognosis of HNSCC and hold potential as effective therapies for various cancers in the future [25,26]. Yet, there is a notable gap in prognostic studies exploring genes associated with cuproptosis and ferroptosis in HNSCC. Therefore, focusing on the potential prognostic studies based on the connections of cuproptosis and ferroptosis in HNSCC is crucial. 

Although the critical roles of cuproptosis and ferroptosis in the development of HNSCC are well established, few studies have investigated the role of genes co-associated with both in HNSCC prognosis. In this study, we identified 12 prognostic genes (the NAD(P)H: quinone oxidoreductase 1 (NQO1), peroxiredoxin 6 (PRDX6), heat shock protein family A member 5 (HSPA5), voltage-dependent anion channel 2 (VDAC2), GABA type A receptor-associated protein-like protein 2 (GABARAPL2), ATP5MC3, autophagy-related gene 5 (ATG5), G6PD, AURKA, CDKN2A, mitogen-activated protein kinase (MAPK9), and Caveolin1 (CAV1)) associated with both cuproptosis and ferroptosis in HNSCC. NQO1 was found to be associated with lower overall survival in HNSCC patients [27]. Consistent with our results, HSPA5 was confirmed through COX regression analysis to be linked to poor prognosis in HNSCC patients [28]. GABARAPL2 and MAPK9 were associated with autophagy and showed promise as prognostic markers for HNSCC patients [29,30]. Additionally, high levels of ATG5 expression may be associated with low OS in HNSCC patients [29]. In total, 105 patients with HNSCC had high expression of G6PD, which may correlate with lymphatic metastasis and prognosis [21]. CDKN2A deletion mutation is related to the TMB of HNSCC [22]. Immunohistochemical results from 173 HNSCC patients suggest that the overexpression of CAV1 may contribute to local recurrence after radiotherapy through the CAV1/Epiregulin (EREG)/yes-associated protein (YAP) pathway [31]. The relationship between several other genes (PRDX6, VDAC2, GABARAPL2, ATP5MC3) and HNSCC has yet to be elucidated and requires further exploration. Appendix A shows the RNA expression profiles of genes for which we constructed risk prognostic models.

We developed a risk-predictive model utilizing the 12 cuproptosis-associated ferroptosis genes in HNSCC. Through single-factor COX regression, multifactor COX regression, and validation using ROC and PCA analysis, our findings indicated that the model was independent of clinical factors and held promise as a prognostic factor in HNSCC. External validation conducted using the GSE41613 dataset further validated the efficacy of our prognostic model in distinguishing between high- and low-risk HNSCC patients. Additionally, we utilized nomogram plots to predict patients’ 1-, 3-, and 5-year survival rates, revealing the robust predictive performance of the model. These results underscore the potential of a prognostic model incorporating genes associated with cuproptosis and ferroptosis to serve as a valuable tool for prognosis, diagnosis, and treatment planning in HNSCC.

In addition, we explored the immune characteristics, biological functions, and drug sensitivity of the prognostic model. In previous studies, it has been demonstrated that patients with high levels of CD8^+^ T cell infiltration may portend a good prognosis [32]. Our findings correspond with them, with high levels of CD8^+^ T cell infiltration in the low-risk group population [33]. The tumor immune dysfunction and rejection (TIDE) is consistent with a tumor immune escape profile and may predict the efficacy of immunosuppressive therapy. Our study is consistent with the results of several studies that showed lower TIDE scores in the high-risk group population, which may be suggestive of the immune escape of tumor cells and lead to poor prognosis. However, there are also some studies that are contrary to our results. For example, one study found that both low TIDE scores and the immune infiltration of CD8^+^ T cells were associated with a better ICB response [34]. Our study corroborates the results of other parts of the studies, and the relationship between the three is complex enough to warrant further investigation in subsequent studies. In conclusion, we simply observed that patients in the high-risk group had lower TIDE scores and lower CD8^+^ T cell infiltration [35,36,37]. The drug sensitivity results suggest that Tivozanib, Vinorelbine, Tipifarnib, Thapsigargin, Talazoparib, Sorafenib, Salubrinal, Roscovitine and WZ184, THZ21021, and THZ249 may be potential agents for the treatment of patient populations at high and low risk of HNSCC, respectively. However, more in- depth clinical outcomes must be explored and validated.

Among the 12 cuproptosis-associated ferroptosis genes, AURKA belongs to the aurora family of kinases and plays a crucial role in mitosis, spindle assembly, and cytoplasmic division of cells, and is a driver of several cancers [38]. In cancer, abnormal expression or mutations of the AURKA gene are closely associated with the occurrence and development of various cancers, including breast cancer, ovarian cancer, gastric cancer, and others [39]. Therefore, AURKA has garnered widespread attention as a potential target for cancer treatment. Several drugs, such as Alisertib, Barasertib, and Danusertib, have been designed to inhibit the activity of the aurora A kinase, with the hope of achieving efficacy in cancer therapy [40]. Recent studies have also revealed that AURKA is a marker protein regulating ferroptosis in tumors. The regulation of AUKRA may affect multiple signaling pathways in tumors, such as the PI3K/Akt, mTOR, β-catenin/Wnt, and NF-κB pathways, and so on [38]. For example, in breast cancer cell lines, the inhibition of AURKA expression increases drug sensitivity to PI3K pathway inhibitors, which in turn induces apoptosis [41]. In gliomas, the knockdown of AUKRA may destabilize β-catenin, a key factor in the regulation of the Wnt signaling pathway, which in turn inhibits cancer development [42]. In gastric cancer cells, selective inhibition using AUKRA significantly reduced NF-κB activity in human gastric cancer samples and mouse epithelial cells, which in turn inhibited the activation of inflammatory signaling pathways in cancer [43]. In addition, studies by Huimou Chen and others have reported that AURKA induces apoptosis and ferroptosis in Ewing’s sarcoma cells through the NPM1/YAP1 axis [44]. Ophiopogonin B can induce ferroptosis in NSCLC by modulating AURKA [45]. AURKA is also identified as a prognostic ferroptosis-related gene signature in adrenal cortical carcinoma [46,47]. In addition, it has been demonstrated that the upregulation of AURKA expression in patients with HNSCC containing TP53 mutations and HPV is associated with poor prognosis and cisplatin resistance in patients. Using an AURKA inhibitor (Alisertib) may result in spindle defects, G2/M arrest, inhibition of cyclin-dependent kinase 1 phosphorylation, and cell proliferation capacity in FaDu and UNC7 cells [17]. However, few studies related to AURKA in HNSCC have been conducted. Our study revealed that AURKA might be a useful biomarker in HNSCC based on Pearson correlation and COX and LASSO analysis. AURKA may be associated with cuproptosis genes, such as DBT, dihydrolipoamide S-succinyl transferase (DLST), lipoic acid synthase (LIAS), and pyruvate dehydrogenase E1 alpha (PDHA1) in HSNCC, and the potential mechanism of their interactions still needs to be further explored. To further explore the function of AURKA in HNSCC, we designed two siRNAs and transfected them into two HNSCC cells (Cal27 and CNE2). We validated the siRNAs by RT-qPCR and performed subsequent cellular function assays to explore the effects. The study demonstrated that the suppression of AURKA markedly decreased the proliferation and migration capability of Cal27 and CNE2 cells. Our findings suggest that AURKA may be a promising prognostic marker and therapeutic target associated with cuproptosis and ferroptosis in HNSCC.

Our study has several limitations: first, our sample size is relatively small, although we used the sample size in TCGA as well as the GEO database for external validation. However, due to the imperfect sequencing platform and the lack of clinical information on some of the samples, we still lack a large-scale cohort for in-depth validation of our model. This may cause us to have missed some of the potential biomarkers. The relatively small sample size may not be fully representative of the entire HNSCC population. Therefore, in a follow-up study, we will collect a larger clinical sample to validate the generalizability of the prognostic model we constructed. Second, although our study validated the function of AURKA in HNSCC by RT-qPCR and functional assays, its precise molecular regulatory mechanism still needs to be confirmed by further studies. We need more clinical samples to corroborate our findings. Moreover, the regulatory mechanisms of copper death and iron death are very complex. Although we found that AURKA, as a ferroptosis-related gene, may be associated with cuproptosis-related genes such as DBT, DLST, LIAS, and PDHA1, how they interact with each other to influence cancer development needs to be investigated in depth. We hope to construct stable cell lines knocking down and overexpressing these genes, respectively, in our subsequent studies, and to explore and validate the results obtained by molecular biology experiments such as RT-qPCR, Western blotting, co-immunoprecipitation, and dual-luciferase reporter genes at the cellular and animal levels, respectively, which will be carried out in our lengthy subsequent studies. In addition, we need to investigate the relationship between copper death, iron death, and the therapeutic strategy and prognostic assessment of HNSCC in more depth in our future studies. In the future, we will conduct more experimental validation and clinical studies to further elucidate the biological functions and potential regulatory mechanisms of the 12 genes we identified in HNSCC.

In summary, our study presents the inaugural creation of a prognostic model for HNSCC founded on cuproptosis-associated ferroptosis genes. Univariate COX regression analysis, multifactor COX regression analysis, Kaplan–Meier, and ROC in both the TCGA and GEO databases indicated that the prognostic model serves as an independent prognostic predictor. Among the 12 genes utilized in the model, AURKA exhibits promising potential as both a predictor and therapeutic target in HNSCC.

## 4. Materials and Methods

### 4.1. Data Acquisition

For this study, we collected RNA transcript data and clinical information from the TCGA database. We obtained RNA transcript data for 546 patients, including 44 normal samples and 502 tumor samples. After merging and removing duplicates and normal samples, 502 samples were finally included for analysis. For external validation of the risk prognostic model we constructed, we retrieved clinical and transcriptional data from the GSE41613 (GPL570 platform) from the Gene Expression Omnibus (GEO) database, which comprises 97 OSCC samples. 

### 4.2. Identification of Cuproptosis-Associated Ferroptosis Genes in HNSCC

To identify the cuproptosis and ferroptosis genes associated with each other in HNSCC, we collected 19 cuproptosis genes from the published literature and acquired ferroptosis genes from the Ferroptosis Database (FerrDb) database (http://www.zhounan.org/ferrdb/legacy/index.html, accessed on 2 January 2023). We conducted Pearson correlation analysis between the two gene sets, using |Pearson| > 0.3 and *p*-value < 0.001.

### 4.3. Construction of a Predictive Model for HNSCC Based on Cuproptosis-Associated Ferroptosis Genes

To identify prognostic genes associated with ferroptosis and cuproptosis in HNSCC, we performed univariate Cox proportional hazard regression analysis using the survival package (*p* < 0.05). Using the glmnet R package (version 4.3.2-64bit), the predictive signature based on prognosis-related genes was subsequently identified by least absolute shrinkage and selection operator (LASSO) Cox regression analysis. Finally, 12 cuproptosis-associated ferroptosis genes were screened out on the basis of the optimal penalty parameter λ determined by tenfold cross-validation following the minimum criteria. The TCGA dataset was partitioned into sets for training and validation purposes (8:2). The training set constructed the predictive model for cuproptosis-associated ferroptosis genes, and the testing set was used to verify and optimize the predictive model. The risk score can be calculated by the following equation [48]:(1)$$risk\;scores=\sum_{i=1}^{n}coef(Expression\;i)\times expr(Expression\;i)$$

*coef* and *expr* denote the coefficients and the expression of cuproptosis-associated ferroptosis genes, and coef is *β* in the multivariate Cox regression analysis formula. In the multivariate Cox regression analysis model, *β* is the regression coefficient. For genes with a high β value, the danger (risk of death) is higher, so the prognosis is worse.
(2)$$h\left(t\right)=h_{0}\left(t\right)e^{\beta 1x1+\cdots \ldots +\beta nxn}$$

Based on median risk scores, all patients with HNSCC were divided into high-risk and low-risk groups. We then validated the ability of the prognostic model to discriminate between the two groups using PCA and risk score correlation analysis. The ROC curves objectively evaluated the accuracy of the prognostic model. 

Furthermore, a Kaplan–Meier analysis was performed using the survival and survminer software (version 4.3.2-64bit) suites to investigate the potential use of the risk score as a clinical prognosis independent predictor via a comparison of high- and low-risk groups. To assess the prognostic significance of the prognostic model for various clinical characteristics, univariate and multivariate Cox regression analyses were performed. Finally, we created a line plot representing the risk column for HNSCC.

### 4.4. Immune Characteristics and Functional Analysis

Expression data (ESTIMATE) are utilized to estimate the quantity of stromal and immune cells in malignant tumor tissue, thereby evaluating tumor purity [49,50]. The presence of immune cells in high- and low-risk patients is evaluated through the use of the following seven algorithms: TIMER, CIBERSORT, quanTIseq, xCell, MCP-counter, and EPIC [51,52,53,54,55]. In addition, patients’ immune response was evaluated using TISIDB, tumor mutational load (TMB), and TIDE [56,57].

### 4.5. Biological Functional Enrichment Analysis and Drug Sensitivity Analysis

To investigate the biological functional differences between high- and low-risk populations for HNSCC, we conducted functional enrichment analysis of various subtypes using the GSEA and cluster profile packages. KEGG enrichment mainly uses the following web pages: (www.genome.jp/kegg/, accessed on 2 January 2023). Pathways with a *p*-value < 0.05 were considered statistically significant signal pathways. Furthermore, to explore potential drugs for high-risk populations, we assessed the drug sensitivity of cuproptosis-associated ferroptosis genes in HNSCC using the pRRophetic package (version 4.3.2-64bit).

### 4.6. Cell Culture

We obtained two HNSCC cell lines, Cal27 and CNE2, from the Cell Bank of the Institute of Biological Sciences, Chinese Academy of Sciences. The CNE2 cells were cultured in Roswell Park Memorial Institute (RPIM) 1640 medium supplemented with 10% Fetal Bovine Serum (FBS) (Gibco, CA, USA), while the Cal27 cells were cultured in Dulbecco’s Modified Eagle’s Medium (DMEM) medium (Yeasen Biotechnology, Shanghai, China) supplemented with 10% FBS. Both cell lines were grown in a 5% CO_2_ incubator at a temperature of 37 °C. The media were changed every two days.

### 4.7. siRNA Transfection

The functionality was confirmed through the siRNA knockdown of AURKA. The siRNA was synthesized by Beijing Dyna Science Biologicals, and cells were plated and transfected with siRNA-NC and two siRNA-AURKA using lip2000 (Invitrogen, Waltham, MA, USA), according to the manufacturer’s instructions. RT-qPCR and subsequent functional validation were performed 24, 48, or 72 h post-transfection.

### 4.8. RT-qPCR

Total RNA was extracted from the cells employing the Trizol method (Takara, Shiga, Japan). The extracted total RNA was reverse transcribed and subsequently quantified by real-time fluorescence using the Takara RT reagent (Yeasen Biotechnology, Shanghai, China), following the manufacturer’s instructions. The upstream and downstream primer sequences for AURKA are as follows: AURKA(F): GAAGCAATTGCAGGCAACCA; AURKA(R): GAGGGCGACCAATTTCAAAG; GAPDH(F): ACCCAGAAGACTGTGGATGG; and GAPDH(R): TTCAGCTCAGGGATGACCTT. All the results were calculated using the 2^−ΔΔCT^ method after normalizing to GAPDH. All the experiments were repeated three times.

### 4.9. Cell Counting Kit8 Experiment

Cell viability was assessed through the use of the CCK8 (Yeasen Biotechnology, Shanghai, China) experiment. After siRNA transient transfection (including the control), the CNE2 and Cal27 cells were plated into a 96-well plate with approximately 3000 cells in each well, and incubation was continued for 0, 24, 48, 72, and 96 h. Next, each well was seeded with 10 μL CCK8 solution. Subsequently, the incubation time was approximately 12 h. The optical density (OD) was assessed utilizing a microplate reader at a wavelength of 450 nm, following the kit’s guidelines. Each experiment was independently repeated in triplicate.

### 4.10. Wound Healing

After efficient centrifugation, the CNE2 and Cal27 cells (including the control) were seeded into a 24-well plate. Once the cells reached 100% confluence, sterile pipette tips created cell scratch wounds. Cell cultures were incubated in a serum-free medium for 0, 24 and 48 h. All the experiments were performed with triplicate samples. Images were taken using an inverted microscope. The migration distance of the wound was calculated using Image J software (version 1.53).

### 4.11. Statistical Analysis

The statistical analysis was carried out using GraphPad Prism 9. The two sets of data were compared using t-tests. All the presented data are represented by mean values with standard errors of the means. *p* < 0.05 was statistically significant.

## Figures and Tables

**Figure 1 ijms-25-04372-f001:**
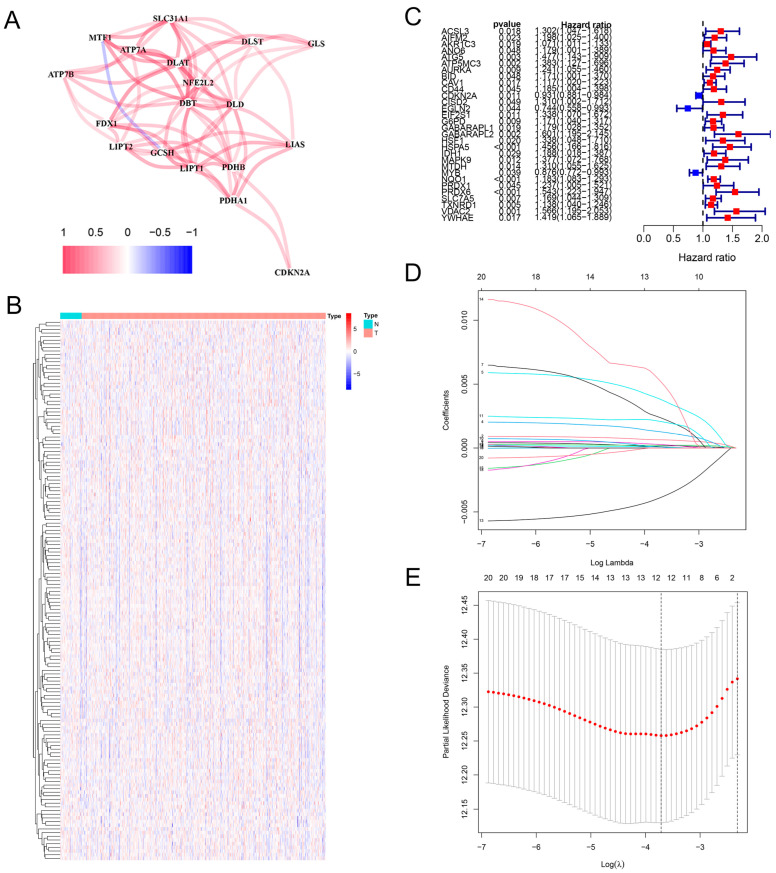
Construction of the risk model based on cuproptosis-associated ferroptosis genes in HNSCC. (**A**) Correlation networks explain the interactions between cuproptosis genes in HNSCC: red represents positive correlations; blue represents negative correlations. (**B**) Clustered heatmap of the expression of 150 cuproptosis-associated ferroptosis genes in HNSCC. (**C**) One-way regression analysis of cuproptosis-associated ferroptosis genes in HNSCC. (**D**,**E**) LASSO regression analysis with cuproptosis-associated ferroptosis genes in HNSCC. Coefficient profiles were drawn based on (log λ) sequences and the value of lambda. Min was defined based on 10-fold cross-validation, where the optimal λ yielded 12 cuproptosis-associated ferroptosis genes.

**Figure 2 ijms-25-04372-f002:**
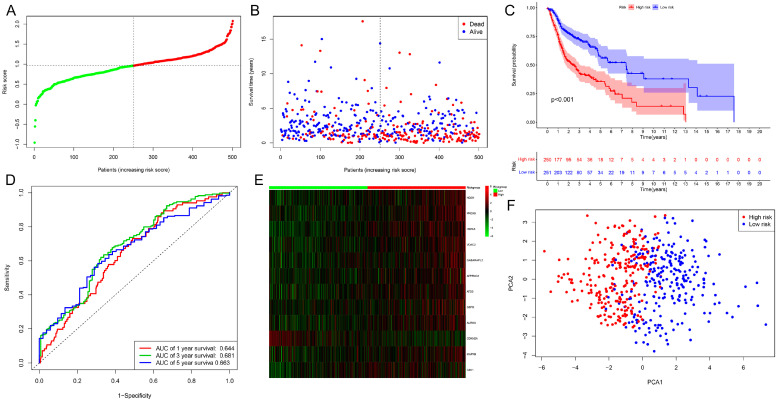
The risk model was constructed based on the TCGA training set. (**A**) The plot of the distribution of patient risk rating points in the TCGA training population. The red curve represents those in the high-risk group and the green represents those in the low-risk group in the TCGA testing population. (**B**) The scatterplot of patient survival status in the TCGA training population. (**C**) The Kaplan–Meier survival plots for patients in the TCGA training set population. (**D**) The ROC curve analysis of predictive performance of assessment risk prognostic model in the TCGA training set population. (**E**) The clustered heatmap of the expression of the twelve cuproptosis-associated ferroptosis genes modeled in two groups of people at high and low risk. (**F**) In the TCGA training set, PCA analyses assess the discriminatory ability of our constructed prognostic model.

**Figure 3 ijms-25-04372-f003:**
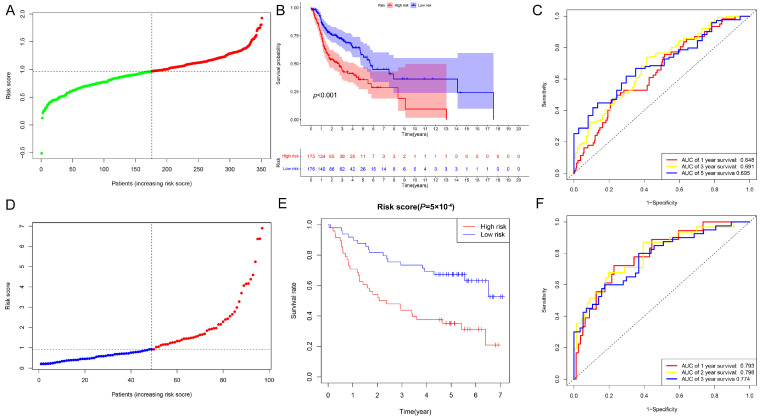
Validation of the risk model in HNSCC. (**A**) The plot of the distribution of patient risk rating points in the TCGA testing population. The red curve represents those in the high-risk group and the green represents those in the low-risk group in the TCGA testing population. (**B**) The Kaplan–Meier survival plots for patients in the TCGA testing set population. (**C**) The ROC curves analysis of predictive performance of assessment risk prognostic model in the TCGA testing set population. (**D**) The plot of the distribution of patient risk rating points in the GEO validation database. The red curve represents those in the high-risk group and the blue represents those in the low-risk group in the GEO validation database. (**E**) The Kaplan–Meier survival plots for patients in the GEO validation database. (**F**) The ROC curves analysis of predictive performance of assessment risk prognostic model in the GEO validation database.

**Figure 4 ijms-25-04372-f004:**
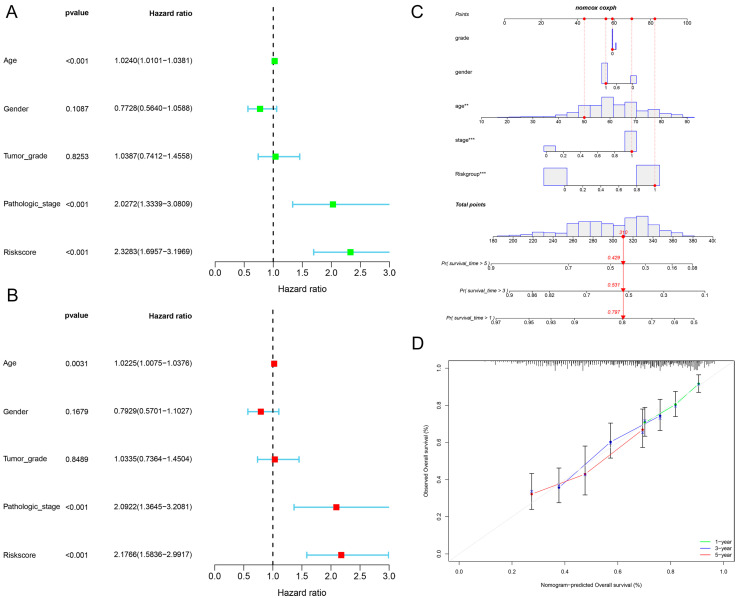
Validation of the predictive performance of our risk prognostic model constructed on the basis of 12 cuproptosis-associated ferroptosis genes in HNSCC. (**A**) One-way regression analyses were performed to validate the predictive performance of the risk prognostic model we constructed. (**B**) Multifactorial regression analyses were performed to validate the predictive performance of the risk prognostic model we constructed. (**C**) The prognostic nomogram graph for a given patient was assessed using the risk prognostic model we constructed (** *p*-value < 0.01, *** *p*-value < 0.001). The red numbers in the column line graph represent the overall score and predicted 1-year survival, 3-year survival, and 5-year survival for a given patient, respectively. (**D**) The calibration plot for a given patient was assessed using the risk prognostic model we constructed.

**Figure 5 ijms-25-04372-f005:**
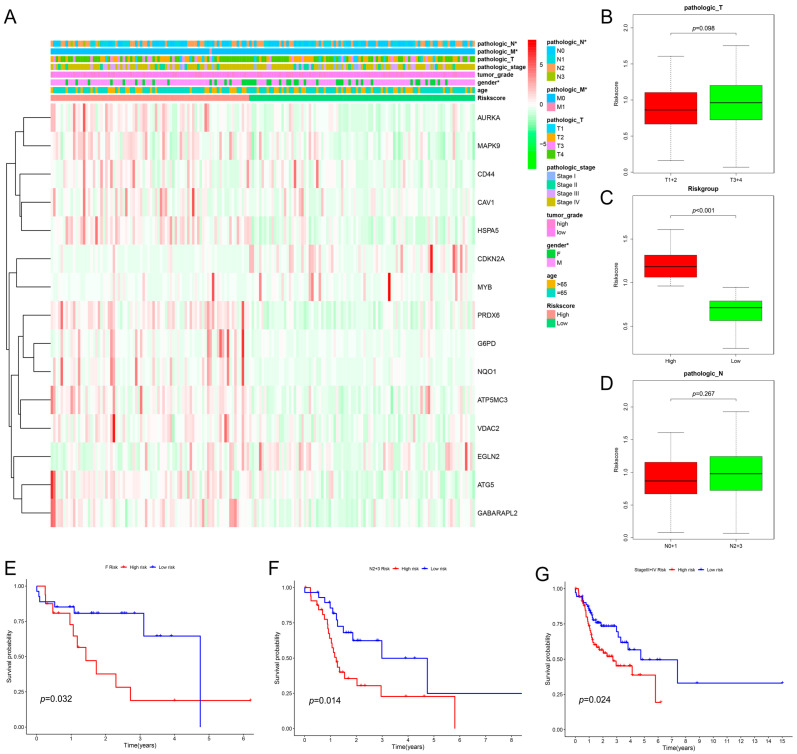
The correlation between the risk model and clinical characteristics. (**A**) Heatmap depicting the relationship between risk model and clinical characteristics. The risk prognostic model correlates with TNM staging. (**B**) T-staging characteristics of the risk prognostic model. (**C**) Risk score grouping characteristics of the risk prognostic model. (**D**) Tumor N grading characteristics of the risk prognostic model. (**E**) Kaplan–Meier survival curve analysis of female patients. (**F**) Kaplan–Meier survival curve analysis of patients with stage N-stage 2–3. (**G**) Kaplan–Meier survival curve analysis of patients with tumor stage 3–4. (**H**) Kaplan–Meier survival curve analysis of patients with tumor T-stage 1–2. (**I**) Kaplan–Meier survival curve analysis of patients with tumor T-stage 3–4. (**J**) Kaplan–Meier survival curve analysis of M0 patients. * *p*-value < 0.05.

**Figure 6 ijms-25-04372-f006:**
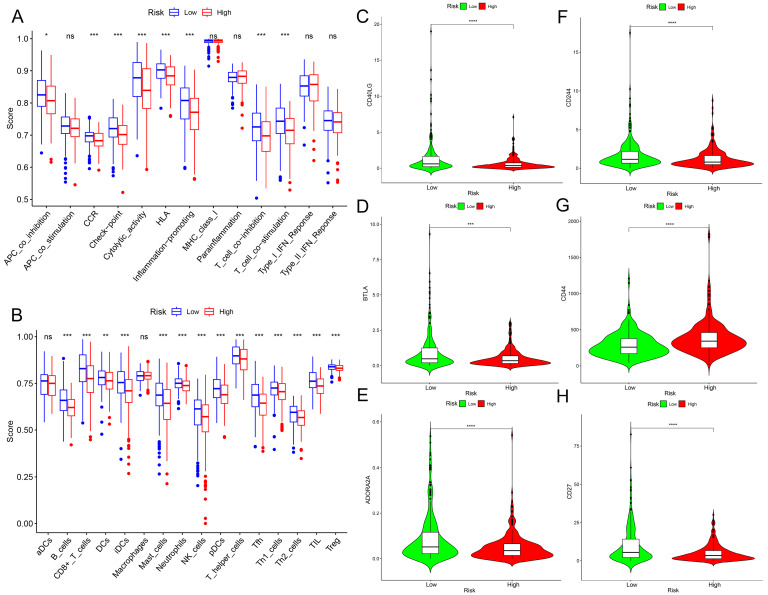
The correlation between the risk model and immunological characteristics. (**A**) The histograms of the immunological function analysis of the two populations in the high-risk and low-risk groups. (**B**) The histograms of the immune cell analysis of the two populations in the high-risk and low-risk groups. (**C**) The results of the analysis of the difference in the immune checkpoint “CD40LG” between the high-risk and low-risk groups. (**D**) The results of the analysis of the difference in the immune checkpoint “BLTA” between the high-risk and low-risk groups. (**E**) The results of the analysis of the difference in the immune checkpoint “ADORO2A” between the high-risk and low-risk groups. (**F**) The results of the difference analysis of immune checkpoint “CD244” between the high-risk and low-risk groups. (**G**) The results of the difference analysis of immune checkpoint “CD44” between the high-risk and low-risk groups. (**H**) The results of the difference analysis of immune checkpoint “CD27” between the high-risk and low-risk groups. * *p*-value < 0.05, ** *p*-value < 0.01, *** *p*-value < 0.001, **** *p*-value < 0.0001, ns: not significant.

**Figure 7 ijms-25-04372-f007:**
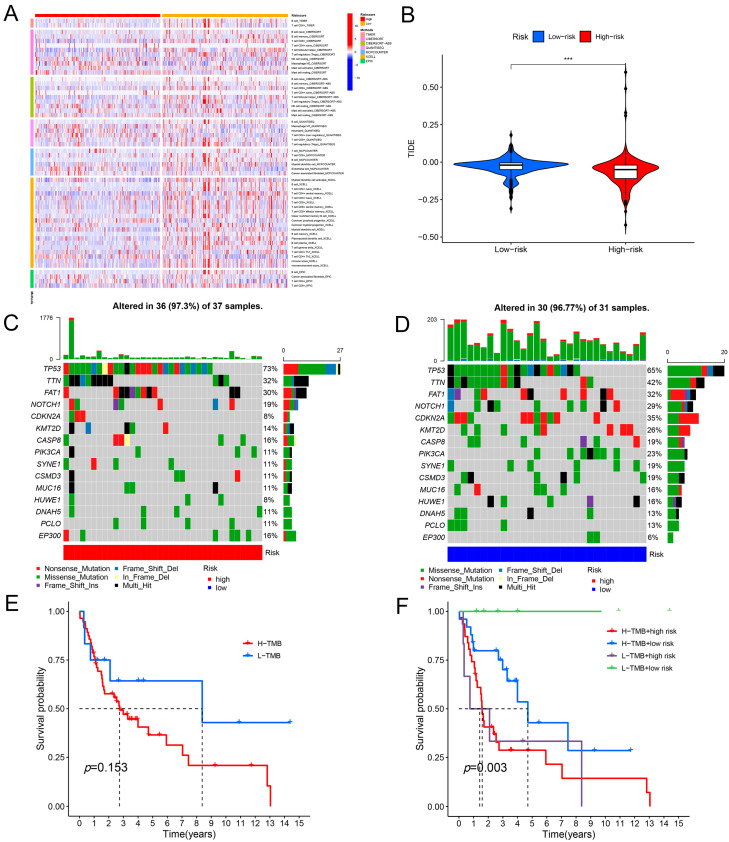
The correlation between the risk model and immunological characteristics. (**A**) Immune infiltration characteristics of the population in the high- and low-risk groups were calculated based on seven algorithms. (**B**) TIDE scores of people in the high- and low-risk groups. (**C**) Characteristics of the first 15 TMB mutations in the population of the high-risk group. (**D**) Characteristics of the first 15 TMB mutations in the population of the low-risk group. (**E**) Results of KM survival analyses for populations with different TMB scores. (**F**) Results of KM survival analysis for different scores in different subgroup populations. *** *p*-value < 0.001.

**Figure 8 ijms-25-04372-f008:**
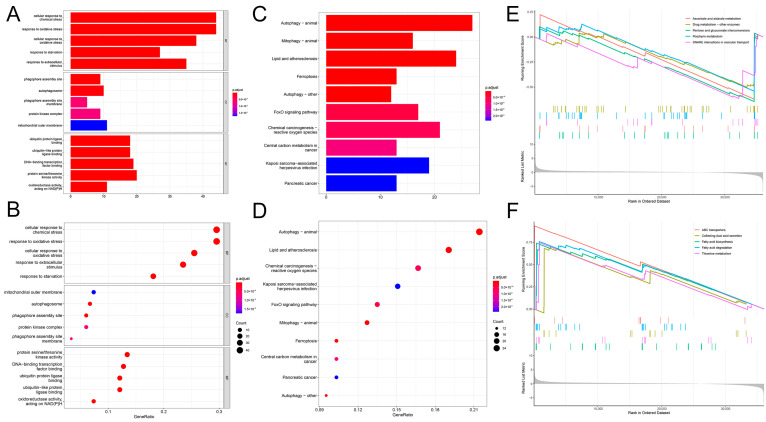
The correlation between the risk model and biological function characteristics. (**A**,**B**) The bar and bubble plots of the results of GO enrichment analyses in the at-risk population. (**C**,**D**) The bar and bubble plots of KEGG enrichment analysis results for the risk population. (**E**,**F**) The GSEA enrichment analysis results for the at-risk population.

**Figure 9 ijms-25-04372-f009:**
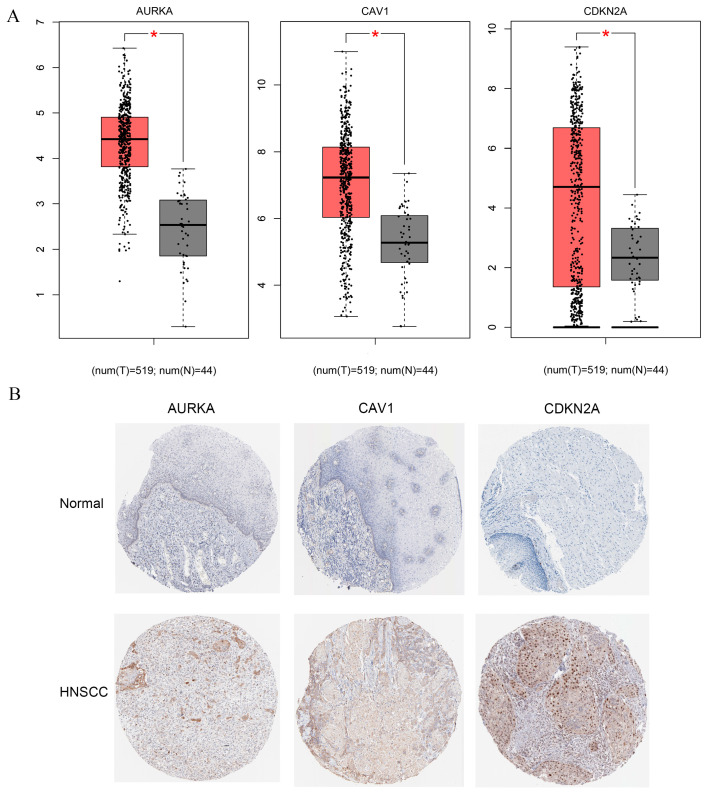
Identification of potential biomarkers in constructed model. (**A**) The RNA expression analysis of AUKRA, CAV1, and CDKN2A in normal tissue and HNSCC samples. * *p*-value < 0.05 (**B**) The analysis of immunohistochemical results of AUKRA, CAV1, and CDKN2A in normal tissue and HNSCC samples.

**Figure 10 ijms-25-04372-f010:**
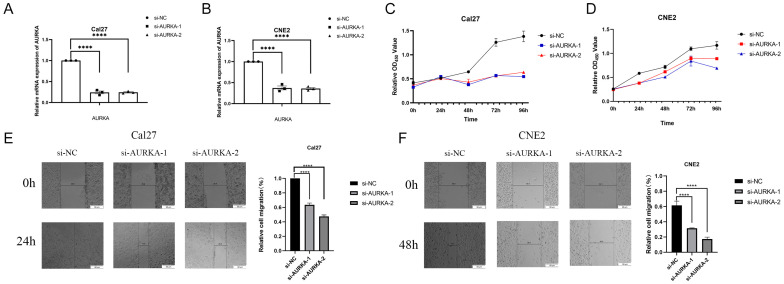
Inhibition of AURKA suppresses the proliferation and migration capabilities of HNSCC cells. (**A**,**B**) RT-qPCR analysis demonstrates the efficiency of AURKA knockdown in Cal27 and CNE2 cell lines. (**C**,**D**) The CCK8 cell proliferation assay after AURKA knockdown in Cal27 and CNE2 cell lines. (**E**,**F**) The cell scratch test results showed that the AURKA knockdown impedes the cellular migration of Cal27 and CNE2 cells (scale bar: 50 μm). The data are presented as relative cell migration (%) at 0, 24, or 48 h. **** *p*-value < 0.0001.

## Data Availability

All the data for this study were obtained from publicly available databases such as TCGA (https://portal.gdc.cancer.gov/, accessed on 2 January 2023) and GEO (https://www.ncbi.nlm.nih.gov/geoprofiles/, accessed on 2 January 2023).

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
