# Peer review of "Identification of AURKA as a Biomarker Associated with Cuproptosis and Ferroptosis in HNSCC"

_ijms, 2024, doi:10.3390/ijms25084372_

Round 1

Reviewer 1 Report

Comments and Suggestions for Authors

The manuscript entitled “Identification of AURKA as a biomarker associated with Cu-proptosis and ferroptosis in HNSCC” discusses a relevant topic. The authors had previously found that of the cuproptosis-associated ferroptosis genes modelled, aurora kinase A (AURKA) was upregulated in HNSCC patients with the tumor suppressor p53 (TP53) mutations and human papillomavirus uninfected (HPV) and could be associated with poor prognosis. Therefore, they intended to investigate the potential prognostic value of AURKA by using functional assays. However, the authors should consider some important points to improve the manuscript.

1.       Please, describe the meaning of GSH, LUAD and BRCA. They are really universally well-known acronyms but it is important to explain their meaning;

2.       Some words appeared typed together in the text.  Please, correct. (line 69 ferroptosisrelated, line 106 predictive predictive signature, line 155 riskprognostic).

3.       The authors stated that they constructed the risk prognosis model for HNSCC to assess the predictive risk of 117 patients based on the cuproptosis-associated ferroptosis genes. Why only 95 patients records were selected? Please, clarify.

4.       Line 410 “Inhibiting AURKA can induce in cancer cells through various pathways [41]”. I think this idea was not clear.

5.       The discussion section was well written. The authors highlighted the limitations of the paper.

Comments on the Quality of English Language

Good quality. Some minor corrections are need.

Author Response

Thank you very much for your review. We very much recognise your comments. We have made changes in response to your comments, and all changes have been highlighted in red in the revised version with the revised version line number where the change was made. Please see the document below for the specific changes you have made. Thank you again for your patience in reviewing this document!

Reviewer 2 Report

Comments and Suggestions for Authors

In this article, the authors picked up Aurora kinase A(AURKA) from cuproptosis and ferroptosis-related genes which gene is more associated with the prognosis of head and neck squamous cell carcinoma (HNSCC) patients in the database.

  1. Most of the data seems to be just pasted charts and tables from database analysis software. At the very least, I would like you to make some effort to make it easier for readers to understand, such as by giving a proper title or by making important titles larger. For example, there are six graphs in Fig. 5C, but it is difficult to understand how they differ.
  2. From 1 to 8 in the figures, the authors have verified the database and verified the correlation with the prognosis of the unique high-risk low-risk grouping and the correlation with clinical data. However, it is difficult to understand how AURKA was narrowed down using the unique prognostic risk model, as it is put in supplemental data. It is better not to make it supplemental data.

Author Response

Dear reviewer,

     Thank you very much for your review. We very much recognise your comments. We have made changes in response to your comments, and all changes have been highlighted in red in the revised version with the revised version line number where the change was made. Please see the document below for the specific changes you have made. Thank you again for your patience in reviewing this document!

      Best wishes!
